# Study on Penetration Mechanism of Shaped-Charge Jet under Dynamic Conditions

**DOI:** 10.3390/ma15207329

**Published:** 2022-10-20

**Authors:** Yizhen Wang, Jianping Yin, Xuepeng Zhang, Jianya Yi

**Affiliations:** College of Electromechanical Engineering, North University of China, Taiyuan 030051, China

**Keywords:** jet, dynamic conditions, virtual origin, penetration mechanism

## Abstract

Aiming at the dynamic penetration process of a shaped-charge jet, we proposed a mathematical model for the penetration of a jet under dynamical conditions based on the theory of virtual origin and the Bernoulli equation taking into account the jet and target intensities. The dynamic penetration process of the jet was divided according to the penetration channel of the jet into the static target. The dynamic penetration model of the jet based on the unperturbed section and perturbed section was established. The penetration depth variation in the shaped-charge jet vertically penetrating target plates with different moving speeds (150~400 m/s) was analyzed by finite element software. The dynamic penetration model shows that with the increase in the target moving speed, the disturbed time of the jet continuously advances, and the dynamic penetration depth continuously decreases; as the velocity of the target increases, the penetration length of the unperturbed jet decreases and then becomes stable, while the penetration length of the perturbed jet decreases. The results showed that the mathematical model is consistent with the finite element simulation, and that the mathematical model can effectively characterize the penetration depth of the unperturbed and disturbed jet portions, adequately explain the dynamic response behavior of the jet penetrating a moving target, and effectively predict the dynamic penetration depth of the jet under the influence of the target movement.

## 1. Introduction

With the development of military technology, a large number of destructive elements with strong penetration capabilities have emerged to destroy underground military targets reinforced by the enemy. Short- and medium-range air defense and antimissile defense are vital methods in protecting our own targets from attack [1]. How to effectively destroy incoming targets has been the main research direction of relevant scholars at home and abroad. Currently, antimissile countermeasures in various countries rely primarily on high-velocity fragmentation to intercept incoming targets, that is, using kinetic energy to destroy the damage element to interfere with the target, directly penetrate it, destroy its critical components or aerodynamic shape, and cause it to deviate from its established trajectory. Alternately, its internal charge is detonated by kinetic impact, pre-detonating the warhead and thus the antimissile. At present, it is commonly believed that directly detonating the incoming warheads is the best [2].

In recent years, the technology of deep-penetrating warheads has been continuously developed, and the thickness of their shells has been increasing. Meanwhile, alloyed steel with great strength and toughness has been used to increase penetration. The internal charge is also increasing, effectively damaging the target. Traditional fragmentation or kinetic bar warheads produce damage elements that do not penetrate the shell effectively due to their large velocity, thickness, and strength. Even if the shell is penetrated, its internal charge cannot be reliably detonated. This situation has posed a serious challenge to the protection of our strategic locations [3,4]. Therefore, it is necessary to use JET as a damage element to effectively damage the incoming target [5,6,7]. The shaped charge has superior penetration capability, as well as the ability to induce a warhead charge, which can be used for air defense and antimissile missions. The shaped jet uses the Monroe effect to generate elevated temperature and a high-speed shaped jet to damage armored targets [8,9]. As an anti-armor method that has been widely used for a long time, it is used in a variety of anti-armor applications, and there is a relatively well-established theoretical basis for the use of anti-armor in combat missions [10,11,12,13]. The existing research on jet penetration under dynamic conditions mainly focuses on the interaction between shaped-charge jets and explosive reactive armor (ERA). The thickness of the face and back plates of the explosive reactive armor is tiny, so the effects of jet velocity decay and jet break and deflection due to the motion of the target plates are normally ignored when the jet penetrates this thin target dynamically. Currently, the penetration mechanism of shaped-charge jets into a moving target with high-velocity thick walls is poorly investigated. Jia Xin et al. established theoretical models such as the lateral drift velocity of the jet when it is interfered with and the penetration depth of the disturbed jet based on the virtual source point theory and the differential element method [14]. Tian Lili and others found through finite element simulations that the higher the velocity of the moving steel target, the larger the velocity loss after jet penetration. In order to ensure the ability to strike high-speed moving targets, it is necessary to use high explosives as the shaped charge and reasonably design the explosive height [15]. According to the published literature at home and abroad, most of the researchers have conducted simulation research on a shaped-charge jet penetrating reactive armor [16,17,18,19], while the research on the penetration mechanism of a shaped-charge jet under dynamic conditions, especially on the penetration mechanism of thick-walled moving targets, has barely been published. Therefore, in this paper, based on the theory of virtual origin of shaped-charge jets and the analysis of finite element simulations of jet penetration in static conditions, we divide the jet penetration process into two phases: unperturbed and perturbed. A theoretical model of jet dynamical penetration was developed based on the Bernoulli equation taking into account the target and jet strengths. The calculation method of jet dynamic penetration depth is obtained, and the theoretical model is verified by simulation results.

## 2. Typical Penetration Theory

### 2.1. Static Armor-Piercing Theory Based on Virtual Origin

The PER theory shows that for a metal jet, there is always a velocity gradient in the length direction, the jet velocity decreases gradually from beginning to end, and the jet velocity distribution is typically linear [20]. Allison and Vitali [21] proposed that it can be assumed that there is a virtual origin, on which the velocity of each jet micro-element has a linear change in space.

According to the virtual origin theory, the jet micro-elements start from the virtual origin, which is the origin of all jets. The jet velocity is linearly distributed along the jet length, and the jet micro-element velocity does not alter during the movement [22]. Thus, the slope of the line formed by the point on the jet penetration curve and the imaginary origin as a function of time during the jet penetration is the velocity of the microscopic element at the jet head at that moment, as shown by the blue line in the Figure 1. The penetration curve is the red dotted line in the figure, and the slope of a point on this line is the penetration velocity of the jet to the target at the corresponding time, as shown in Figure 1 [23].

This theory has certain accuracy for a general single-cone-shaped liner. The formula for calculating the static penetration depth of a jet given by Allison and Vitali [21] is as follows.
(1)$$P=v_{j}t_{0}{\left(\frac{v_{0}}{v_{j}}\right)}^{(1+\gamma )/\gamma }-z_{0}$$
or
(2)$$P=v_{0}t{\left(\frac{t_{0}}{t}\right)}^{\gamma /(1+\gamma )}-z_{0}$$

According to Formulas (1) and (2)
(3)$$v_{j0}=\frac{v_{0}\cdot z_{0}{}^{\gamma }}{{\left(P(t)+z_{0}\right)}^{\gamma }}$$
where *P* is the penetration depth; *v*_j0_ is the velocity of the head of the jet at the intersection of the projectile and target; *v*_0_ is the velocity of the head of the jet as it contacts the target plate; *t*_0_ is the time at which the jet head moves from the virtual origin to the target surface; *z*_0_ is the distance from the imaginary origin to the target surface; $\gamma =\sqrt{\frac{\rho_{t}}{\rho_{j}}}$, $\rho_{t}$ is the target density and $\rho_{j}$ is the jet density; and *P*(*t*) is a function of the penetration depth *P* of time *t* to the jet.

Regardless of the bullet/target strength, according to the Bernoulli equation:(4)$$\frac{1}{2}\rho_{j}{\left(v_{j}-u\right)}^{2}=\frac{1}{2}\rho_{t}u^{2}$$
where *u* is the penetration velocity. The penetration velocity can be expressed as:(5)$$u=\frac{v_{0}}{1+\gamma }{\left(\frac{t_{0}}{t}\right)}^{\frac{\gamma }{1+\gamma }}$$

Eichelberger thinks that the strength of the target plate and jet have an essential influence on the late process of penetration through experimental measurement, and adds an intensity term in Formula (4) [24]. The penetration velocity *u* of the jet is obtained as a function of the target impedance.
(6)$$u=\frac{1}{1+\gamma }\left[v_{j}-\sqrt{\gamma^{2}v_{j}{}^{2}+(1-\gamma^{2})\frac{2\sigma }{\rho_{j}}}\right]$$
where $\rho =\sigma_{t}-\sigma_{j}$, ***ρ*** is the difference between the plastic deformation impedance of the target and the jet.

### 2.2. Analysis of Jet Aperture under Static Conditions

In this part, the aperture of the jet penetrating the target in the quiescent state is analyzed by numerical simulations. In this paper, a truncated cone-shaped liner with a diameter of 100 mm and a charge length diameter ratio of two is selected, and its cone angle is 40°. The blast height is 100 mm, and its model is shown in Figure 2. For this finite element simulation analysis, we used the R11 LS-DYNA solver purchased by the school and single precision was used. The structural dimensions of the shaped charges and the partition of the model mesh are shown in Figure 3.

The explosives are modeled by the HIGH_EXPLOSIVE_BURN constitutive model with a JWL equation of state. The material choice for the drug-type covers is copper and the target plate is 45 steel, and the JOHNSON_COOK model and the GRUNEISEN equation of state are used for the drug-type covers and target plates, with the parameters given in Table 1 and Table 2.

As shown in Figure 2, the static jet penetration adopts a half model, so in the simulation, symmetrical constraints are imposed on the symmetry plane, and the flow-out boundary is set for the air domain boundary.

In the finite element analysis of the static penetration of a jet into a semi-infinite target, to improve computational efficiency and avoid boundary effects affecting the accuracy of the analysis, a dense grid is partitioned for the part of the target that will be in direct contact with the jet. The mesh is a regular hexahedral cell with an edge length of 0.8 mm, and a larger mesh is used for the edges of the target. The two are connected by a transitional mesh.

When the jet first touches the target plate, the velocity of the micro-element at the jet head decreases, and the particle velocity of the target plate at the contact position remains the same. At this point, the percolating jet element has not yet consumed all the energy and mass, and the jet channel diameter is enlarged by the subsequent jet element boost.

During the pit-opening phase, the target plate near the impact point generates high-speed plastic deformation with a large strain rate. Therefore, in the “three high zones”, the target plate near the jet channel has local hardening, especially in the pit-opening stage. Later, as the jet head velocity decreases, the hardening of the target plate gradually decreases, and the hardening of the target plate in the quasi-steady region is significantly lower than in the pit-opening phase. For semi-infinite targets, jets accumulate at the bottom of the hole during the termination phase. Due to the high temperature of the jet, the target plate in contact with it is tempered, which further reduces the intensity of the target plate in this part of the jet, and its hardness is lower than in the quasi-steady region.

At the beginning of the pit-opening phase, the diameter of the jet flowing through the surface of the ejecta target is about 0.66 cm. The jet produces a hole in the target surface with a diameter of about 2–2.5 times the jet diameter, which is about 1.47 cm. During the penetration process in this phase, the energy of the jet micro-element is large and the impedance of the target plate to the jet is relatively small. The diameter of the counterbore produced by the jet gradually increases, as shown in Figure 4 (the scale units in the figure are centimeters and similarly hereinafter), because the subsequent jet micro-element continuously compresses the previous jet micro-element in the radial direction of the jet.

After that, the energy of each jet micro-element decreases continuously with the presence of the velocity gradient due to the decrease in the velocity of the subsequent jet micro-element and the relative decrease in the pressure at the contact position of the jet target plate. As a result, the remaining energy after the axial penetration of the jet is completed becomes smaller and smaller during the penetration of the target slab.

In addition, due to the severe plastic deformation of the target plate during the hole opening, a local hardening of the target plate is induced, which also increases the resistance of the target plate to the radial extension of the jet. Under the combination of the above two effects, the radial re-emission diameter of the jet decreases, as shown in Figure 5. Therefore, there is an extreme value of the jet reaming diameter during the pit-opening phase.

Since the hole digging process only accounts for a limited part of the total jet penetration process, the subsequent penetration enters the quasi-steady stage, and the energy gradient of the jet in this part is relatively slow and the residual energy of the jet micro-element in this part mainly remains stable after the completion of the axial penetration. Moreover, the amplitude of the material hardening in the quasi-steady region is inferior to that in the hole-opening phase, so that the aperture opened by the hole-expanding jet is mainly non-shifting with time and remains stable, as shown in Figure 6.

In the quasi-steady phase, the diameter of the counterbore produced by the jet penetrating the target plate is about 0.936 cm, and the diameter of the jet at the corresponding position is about 0.33 mm, with an aperture about 2.5–3.0 times the diameter of the jet.

Thus, for a typical jet penetrating a homogeneous target, the reduction in the recoiling diameter is mainly present in the intermediate to quasi-steady phase of the pit-opening phase. Fundamentally, it first increases, then decreases and finally tends to stabilize. This is consistent with the equation of penetration and crater growth by a shaped-charge jet under the influence of a shock wave [25].

## 3. Dynamic Penetration Analysis of Jet

### 3.1. Determination of Interference Time

The target slab has a velocity *v*_t_ perpendicular to the penetration direction based on the theory of virtual origin and the jet penetration hole diameter law.

For the interference of the dynamic target plate to the jet, due to the strong role of jet reaming in the pit-opening stage, which provides favorable conditions for jet penetration, and due to the small opening of the jet on the surface of the target plate, and that in the early stage of the pit-opening stage the diameter of the reaming is constantly increasing, under dynamic conditions, the interference of target plate to jet appears near the surface of target plate. Then, the gap between the two, Δ*r*, can be expressed as:(7)$$\Delta r=R_{b}-r_{j}$$
where *r*_b_ is the channel radius near the surface of the target plate, and *r*_j_ is the jet radius at the same position.

Then, the disturbed time *t*_h_ can be expressed as
(8)$$t_{h}=\frac{\Delta r}{v_{t}}$$
where *v*_t_ is the velocity of the dynamic target plate.

Under the same operating conditions, when the parameters of the liner and the charge are in agreement, the jet formed by the liner is predominantly coherent, and the jet channel generated by its penetration into the static target is similar. Therefore, the interference time can be calculated based on the condition of the jet channel in static conditions.

### 3.2. Penetration Analysis of Disturbed Jet

Under the same operating conditions, when the parameters of the liner and the charge are in agreement, the jet formed by the liner is predominantly coherent, and the jet channel generated by its penetration into the static target is similar. Therefore, the interference time can be calculated based on the condition of the jet channel in static conditions. Assuming that the velocity distribution of the jet along the axis is linear, the velocity of the jet micro-element near the surface of the target slab can be obtained from Equations (2) and (3) as follows.
(9)$$v_{\mathrm{jh}}=\frac{z_{0}\cdot v_{j0}{|}_{t=t_{h}}}{P{|}_{t=t_{h}}+z_{0}}$$
where *v*_jh_ is the velocity of the jet micro-element near the surface of the target slab; $v_{j0}{|}_{t=t_{h}}$ is the velocity of the jet head at the *t*_h_ moment; and $P{|}_{t=t_{h}}$ is the penetration depth at the *t*_h_ moment.

During the interference process, the fluid is assumed to be incompressible and inviscid. After the interferometric time, the pre-*v*_jh_ jet micro-elements have significant energy. After the completion of axial penetration, there is still sufficient energy to expand the target plate, so that Δ*r* is greater than zero to overcome the influence of target movement on jet micro-element penetration; the micro-element of JET after *v*_jh_ will be firstly affected by the target plate movement due to its relatively little energy. The velocity of the jet itself in the direction of motion of the target slab is negligible. It is assumed that the target slab will collide with the jet micro-element where the *v*_jh_ is located out of the jet axis as a whole, and the post-*v*_jh_ jet micro-element can be regarded as re-penetrating the target slab.

Compared to the unperturbed part of the head, the velocity of the perturbed jet profile has less kinetic energy and the diameter of the jet profile is relatively larger. The position interfered by the movement of the target plate is mainly concentrated in the area from the near-surface of the target plate to the end of the pit-opening stage *δ* inside.

Then, the disturbed jet part is equivalent to the head velocity *v*_jh_, and the thickness of the jet section pair with linear velocity gradient is *δ*. The interference part of the hole penetrates and then continues to flow to the bottom of the hole. After the penetration of the jamming part is completed, the remnant jet head velocity *v*_ju_ of the jet in this section is:(10)$$v_{\mathrm{ju}}=\frac{v_{\mathrm{jh}}\cdot z_{h}{}^{\gamma }}{(\delta +z_{h}{)}^{\gamma }}$$
where *v*_ju_ is the residual head velocity; *v*_jh_ is the jet velocity at the radial interference position of the target at *t*_h_; *z*_h_ is the distance from the imaginary origin to the *t*_h_ moment of motion of the microscopic elements of the head of the jet; and *δ* is the distance from the near surface of the target plate to the end of the pit-opening stage.

For the jet in the perturbed part, it is necessary to find the part that still contributes to the tunneling bottom penetration. Therefore, the length of the jet section from the effective jet velocity *v*_jk_ to the residual head velocity *v*_ju_ of the disturbed jet Δ*l* can be expressed as:(11)$$\Delta l=(1-\frac{v_{\mathrm{jk}}}{v_{\mathrm{ju}}})[z_{0}+P{|}_{t=t_{h}}+\delta ]$$
where Δ*l* is the length of the jet section from the effective jet velocity to the residual head velocity of the disturbed jet; *v*_jk_ is the effective jet velocity threshold:(12)$$v_{\mathrm{jk}}=v_{t}{}^{\frac{1}{1+\gamma }}+\frac{\gamma }{1+\gamma }v_{\mathrm{ju}}$$

Considering that the energy of the perturbed jet cross section is already predominantly consumed when it overcomes the perturbation and flows to the hole bottom, it is necessary to consider the effect of the jet and target strengths on the jet penetration. The penetration velocity *u*_r_ of the perturbed part is written according to Equation (6):(13)$$u_{r}=\frac{1}{1+\gamma }\left[v_{\mathrm{ju}}-\sqrt{\gamma^{2}v_{\mathrm{ju}}{}^{2}+(1-\gamma^{2})\frac{2\sigma }{\rho_{j}}}\right]$$
where *u*_r_ is the penetration velocity of the perturbed part.

Since the velocity gradient of the jet in the disturbed section is relatively slow, *v*_ju_ can be considered as the velocity of the jet in this section, then the penetration depth of the disturbed section to the hole bottom Δ*P*_r_ is expressed as:(14)$$\Delta P_{r}=\frac{u_{r}\cdot \Delta l}{v_{\mathrm{ju}}-u}$$
where Δ*P*_r_ is the penetration depth of the disturbed section.

### 3.3. Penetration Analysis of Undisturbed Jet

This part of the jet has already penetrated the target before the time of the perturbation. During the penetration of the jet into the target, it is the head part of the jet that touches the target. The energy carried by the jet is strong and the impact of the target slab on the jet is slight. The average velocity of the head and tail of the jet in this section is taken to be the velocity of the jet element in this section:(15)$$v_{\mathrm{jw}}=\frac{v_{j0}{|}_{t=t_{h}}+v_{\mathrm{jh}}}{2}$$
where *v*_jw_ is the velocity of the undisturbed jet; $v_{j0}{|}_{t=t_{h}}$ is the velocity of the jet head at the moment of *t*_h_; and *v*_jh_ is the velocity of jet at disturbed location at the moment of *t*_h_.

In order to better fit the actual situation, considering the influence of target plate and jet strength on the penetration process of the undisturbed jet, the penetration velocity *u*_w_ and penetration depth Δ*P*_w_ of undisturbed jet can be obtained according to Formulas (6) and (13)–(15):(16)$$u_{w}=\frac{1}{1+\gamma }\left[v_{\mathrm{jw}}-\sqrt{\gamma^{2}v_{\mathrm{jw}}{}^{2}+(1-\gamma^{2})\frac{2\sigma }{\rho_{j}}}\right]\;$$
(17)$$\Delta P_{w}=\frac{u_{r}\cdot P{|}_{t=t_{h}}}{v_{\mathrm{ju}}-u_{w}}$$
where *u*_w_ is the penetration velocity of undisturbed jet; Δ*P*_w_ is the penetration depth of undisturbed jet.

Assuming that each section of the jet is ideally attached to the hole after the completion of radial penetration, and the accumulation of the jet at the hole bottom is negligible, the total penetration depth *P* of the jet to the moving target can be considered as the sum of the penetration depth of the disturbed section jet and the undisturbed section jet.
(18)$$P=\Delta P_{r}+\Delta P_{w}$$
where *P* is the final dynamic penetration depth of the jet.

## 4. Finite Element Simulation Analysis and Verification

### 4.1. Model Establishment

The LS-DYNA finite element analysis software was used to establish the three-dimensional model of jet penetration into the dynamic target plate in order to obtain the parameters needed for theoretical calculation of jet head and tail velocity, velocity gradient, and jet diameter when the jet stretched to the surface of the target plate. The arbitrary Lagrangian Euler algorithm was used for explosive and shaped-charge liners, and the Lagrangian algorithm for target plates. Numerical simulations were performed between them via a fluid–solid coupling algorithm. The element grids of each section are hexahedral elements, and the explosive, shaped-charge liner, and air domain are common nodal Eulerian grids. The model diagram is shown in Figure 7.

The material model and equations of state used for the numerical simulations are the same as those used in the analysis of the jet penetration aperture in static conditions in Section 1. In the dynamic penetration model, the symmetry constraint is also set on the symmetry plane of the half model, and the flow-out boundary is set on the air domain boundary, so that all substances can flow out freely. Consistent with the grid partition of the static penetration model, the dynamic target is also meshed with a transitional grid structure. The portion where the possible interaction with the jet in the target plate is divided into a regular hexahedron grid with a side length of 0.1 cm and distant portion is meshed in 0.3 cm × 0.3 cm × 0.16 cm.

### 4.2. Finite Element Simulation

In order to further verify, analyze, and study the penetration mechanism of the jet under dynamic conditions, under the condition of maintaining the blasting height and the same material parameters, the horizontal right velocity was applied to the target plate, which is 150 m/s, 200 m/s, 250 m/s, 300 m/s, 350 m/s, and 400 m/s. Combined with numerical simulations, the penetration of the jet into the target slab at different moving velocities was analyzed and the influence law of various factors was explored. The results of the numerical simulations are shown in Figure 8.

In this simulation, the blast height of the shaped-charge warhead is the same as the diameter of charge, as the blast height is short. The jet does not break in the static armor-piercing process, and the jet penetrates the target continuously. In the process of dynamical penetration, the jet before the target interferes with the jet is also continuous, so it is also a continuous jet penetration process. It can be seen from Figure 7 that when the target plate speed is low, the jet will be destabilized later, and most of the jet that can penetrate has entered the duct, but it is interfered with by the lateral movement of the target plate, and squeezed and collided with the target plate duct, which affects the penetration ability of the jet. The time for the jet to intervene continues to advance as the velocity of the moving target plate increases. A growing number of armor-piercing jets are directly disrupted by the transverse motion of the target plate and fail to enter the penetration channel from the unperturbed jet to the target plate. Instead, the target plate needs to be re-penetrated, bending and stacking occur near the surface of the target plate, and then fracture and secondary collisions with the hole walls occur during the continued flow to the bottom of the pit. The penetration depth of the jet is clearly reduced.

### 4.3. Comparative Analysis of Theory and Simulation

Following the theoretical model developed in Section 2, the penetration depth of the jet under the corresponding dynamical conditions is obtained by calculating the relative errors between five sets of numerical simulation cases. See Table 3 and Figure 9 for a comparison between the penetration depth from numerical simulations of jets and the depth calculated theoretically.

As can be seen from Table 3, the error between the armor-breaking depth calculated by the theoretical model and the simulation results is minor. This method can be used to estimate the armor-breaking depth of a jet penetrating a moving target. As can be seen from Figure 9, the dynamical penetration depth of the jet into the target decreases with increasing velocity of the target.

In order to further study the dynamic penetration mechanism of the jet, according to the dynamic penetration model established previously, the penetration length *P*_w_ of the undisturbed section and the penetration length *P*_r_ of the disturbed section are extracted, and their proportions and changing rules in the dynamic penetration of the jet are analyzed.

As can be seen in Figure 10, the decay of the penetration length *P*_w_ of the unperturbed jet has a tendency to decrease significantly at first, and then to moderate as the velocity of the target is increased. Combined with the dynamic penetration model and numerical simulation analysis, the interference time keeps advancing as the velocity of the target increases, leading to fewer and fewer jets that can penetrate the target before the interference time. Thus, the penetration depth of the unperturbed jet profile decreases. Due to the elevated velocity of the jet head and the tremendous amount of energy it carries, the diameter of the target slab is relatively large during the opening phase. During the dynamic penetration, a part of the jet can avoid being disturbed by the motion of the target plate under the reaming of the head jet micro-element, so that it can follow the head jet micro-element to penetrate further into the target plate. As a result, this part of the jet is able to overcome the effect of the target motion to a certain extent and maintain the penetration capability of the target.

The penetration depth of the perturbed jet profile initially increases slightly and then decreases. This is because when the moving speed of the target plate is low (*v*_t_ ≤ 200 m/s), the effective jet velocity threshold *v*_jk_ is also low, and the disturbed jet flowing to the bottom of the jet channel still has a certain penetration ability after the interference time, which makes the penetration ability of the disturbed jet increase on the whole, but because the penetration ability of this part is relatively weak, the penetration ability of the disturbed jet is not significantly improved. When the target plate speed is further increased (*v*_t_ ≥ 300 m/s), because the effective jet velocity threshold *v*_jk_ is increased, the jet that can effectively penetrate the bottom of the jet crater is continuously reduced, and the penetration length of the disturbed jet is significantly reduced.

Combining the regularity of the penetration lengths of the perturbed and unperturbed profiles, the ratio of the two in the total dynamical penetration depth of the jet is plotted in Figure 11.

Figure 11 shows that the ratio of the unperturbed jet to the overall penetration depth definitely decreases at first and then rises as the velocity of the target is increased. The penetration depth ratio of the perturbed jet is reversed. Based on the analysis of the dynamic penetration mechanism, it can be seen that when the moving speed of the target plate continues to increase, the interference time will advance, resulting in fewer and fewer jets in the undisturbed section, which directly affects the proportion of this part of the jet in the total penetration distance. At the same time, the penetration distance of the perturbed jet increases proportionally to the total dynamical penetration. However, as the target speed continues to increase, when the target speed is greater than 300 m/s, the effective length of the disturbed section jet continues to decrease due to the increase in the random moving speed of the effective jet velocity threshold, which makes the penetration ability of this part of the jet to the bottom of the jet channel begin to weaken. It can overcome the interference of the target slab and gradually reduce the effective part of the penetration that completes the bottom of the jet channel, such that the penetration length of the perturbed section of the jet decreases in proportion to the total penetration length.

However, the unperturbed profile of the jet has a higher energy due to its position at the jet head, which has the effect of expanding the jet channel. This is the reason why the subsequent part of the jet removes the effect of the motion of the target slab within a certain range, allowing the decay of the penetration length of the unperturbed part of the jet to begin to slow down. In addition, the penetration length of the perturbed jet decreases continuously, such that the fraction of the unperturbed jet in the overall fraction starts to rise for target velocities larger than 300 m/s.

The perturbed jet cross section does not fluctuate significantly at early times, and the unperturbed jet cross section continues to decrease in penetration length according to the overall penetration depth variation in two sections of jets. The penetration length of the perturbed jet profile starts to decrease, while the penetration length of the unperturbed jet profile tends to stabilize for target slab motion velocities larger than 300 m/s. Overall, the penetration depth of the jet into the target decreases with increasing velocity of the target.

## 5. Conclusions

In this paper, we established a mathematical model of the dynamic penetration mechanism of a shaped-charge jet based on the virtual origin theory and Bernoulli equation considering the strength of the jet and target. Combined with the formation of the jet channel in the process of jet static penetration, the theoretical model was verified with numerical simulation. We studied the variational law of the dynamic penetration of shaped-charged jets based on the mathematical model developed to characterize the dynamical penetration behavior of perturbed/undisturbed jets. The main conclusions are as follows:When the velocity of the target is less than 300 m/s, the penetration ability of the disturbed jet has no obvious shift and the penetration ability of the undisturbed jet decreases linearly. When the speed of the target exceeds 300 m/s, the penetration length of the disturbed jet decreases linearly while the undisturbed jet tends to be stable.The contribution of the undisturbed jet to the overall penetration of the jet takes 300 m/s target velocity as the inflection point, showing a parabola pattern of decreasing and then increasing; the contribution of the jet in the disturbed section takes the same target velocity as the inflection point, but the shift is opposite to that of the jet in the undisturbed section.The relative error between the established mathematical model and the numerical simulations is less than 10 per cent, which is in agreement with the numerical simulations, compared to the simulations at six different groups of target velocities. We verified that the established mathematical model can reliably predict the penetration depth of the jet under dynamical conditions.

## 6. Discussion

Authors should discuss the results and how they can be interpreted from the perspective of previous studies and of the working hypotheses. The findings and their implications should be discussed in the broadest context possible. Future research directions may also be highlighted.

## Figures and Tables

**Figure 1 materials-15-07329-f001:**
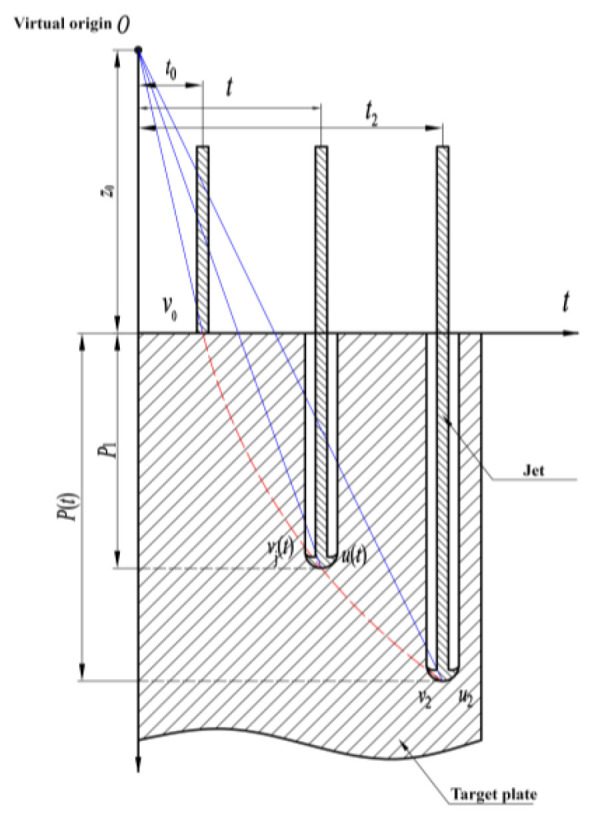
Jet penetration process diagram of virtual origin theory.

**Figure 2 materials-15-07329-f002:**
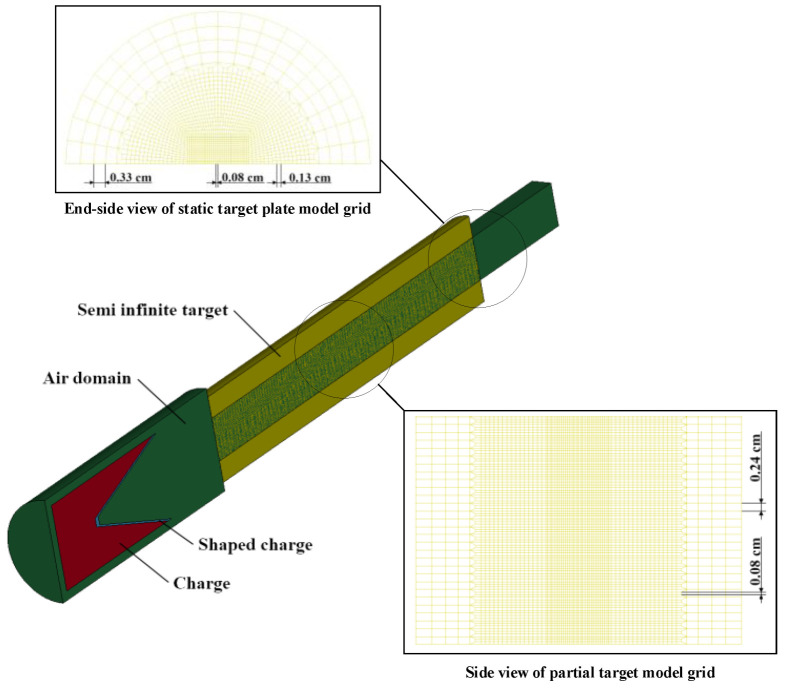
Schematic diagram of jet static penetration.

**Figure 3 materials-15-07329-f003:**
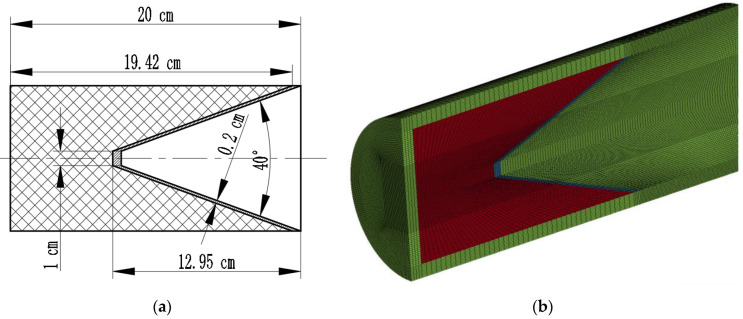
Dimension and grid diagrams of shaped jet. (**a**) Dimensional sketch of shaped jet; (**b**) grid generation of shaped jet.

**Figure 4 materials-15-07329-f004:**
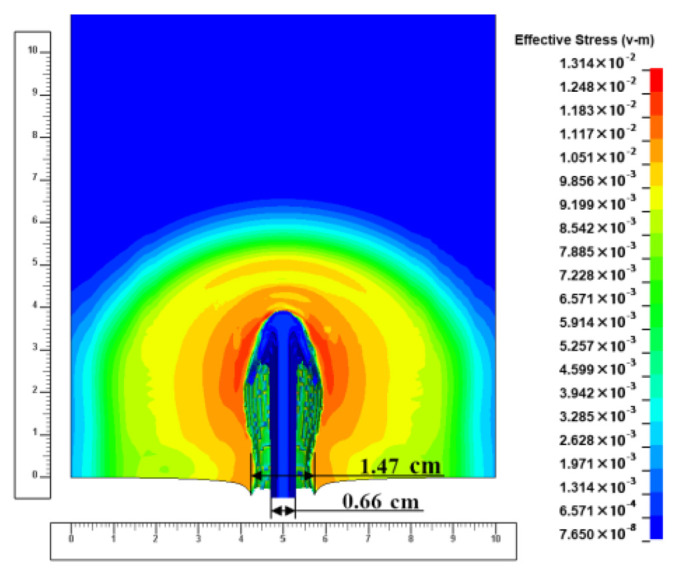
Schematic diagram of the ejecta aperture at the early stage of pit opening.

**Figure 5 materials-15-07329-f005:**
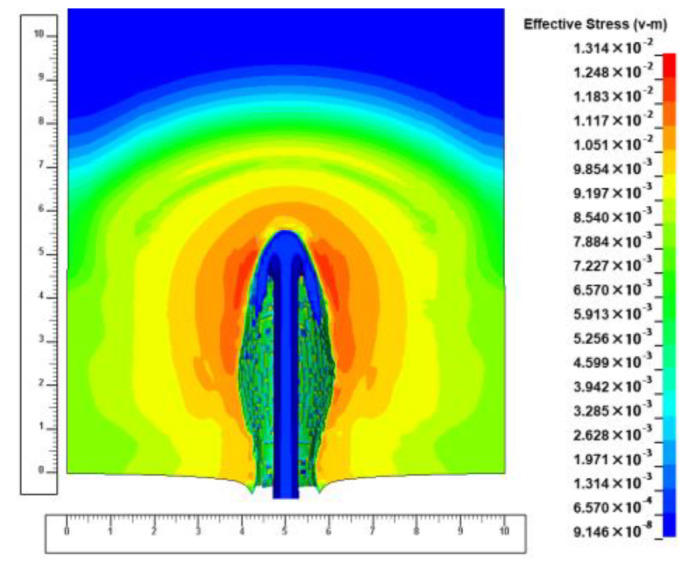
Schematic diagram of the jet aperture at the later stages of the pit-opening.

**Figure 6 materials-15-07329-f006:**
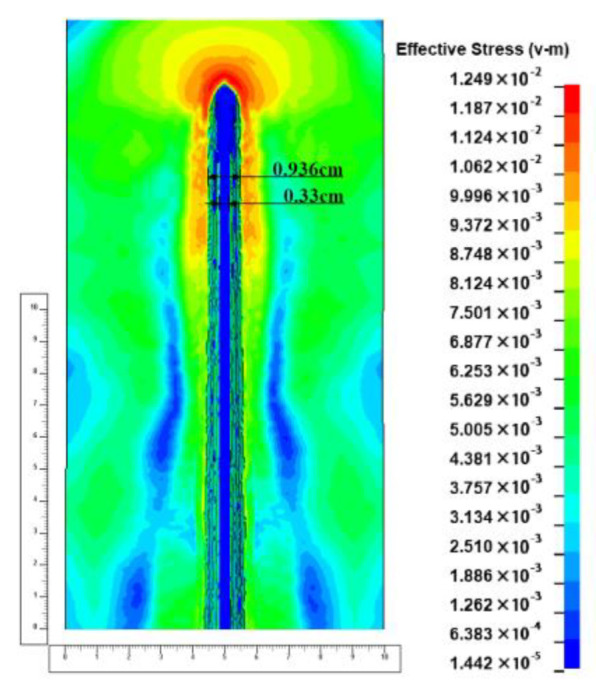
Schematic diagram of the jet aperture during the quasi-stationary phase of jet penetration.

**Figure 7 materials-15-07329-f007:**
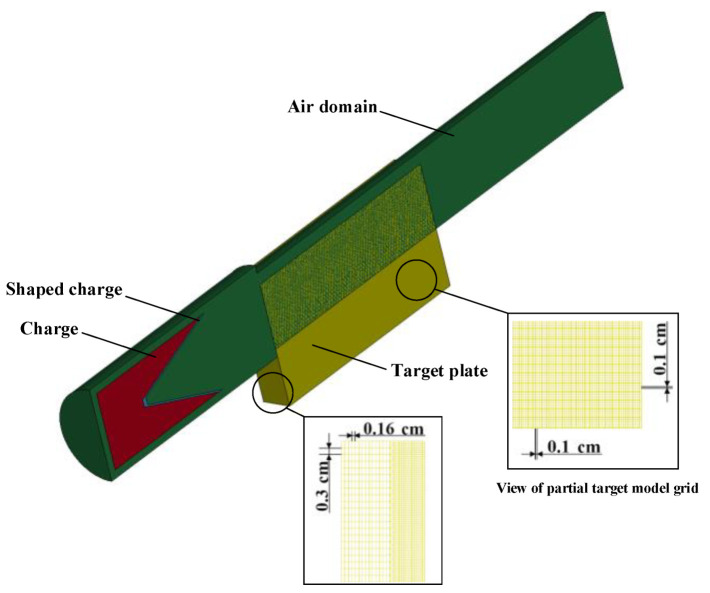
Schematic representation of the numerical simulation model.

**Figure 8 materials-15-07329-f008:**
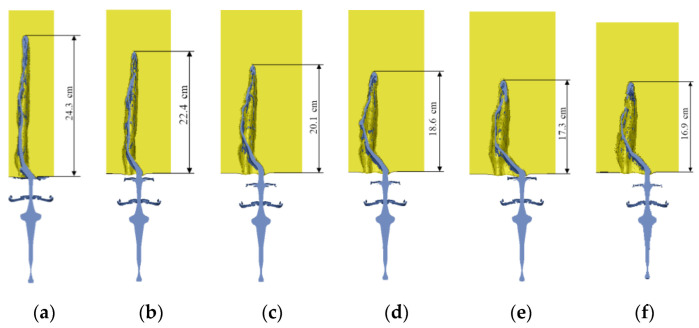
Penetration results at different target velocities: (**a**) *v*_t_ = 150 m/s; (**b**) *v*_t_ = 200 m/s; (**c**) *v*_t_ = 250 m/s; (**d**) *v*_t_ = 300 m/s; (**e**) *v*_t_ = 350 m/s; (**f**) *v*_t_ = 400 m/s.

**Figure 9 materials-15-07329-f009:**
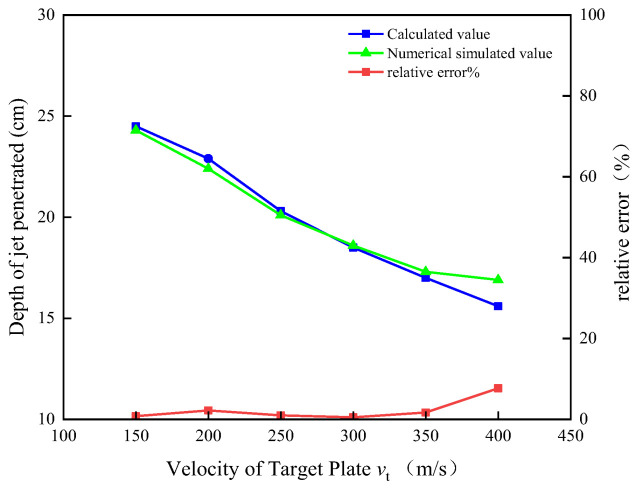
Comparison of jet penetration depth between theoretical calculations and numerical simulations.

**Figure 10 materials-15-07329-f010:**
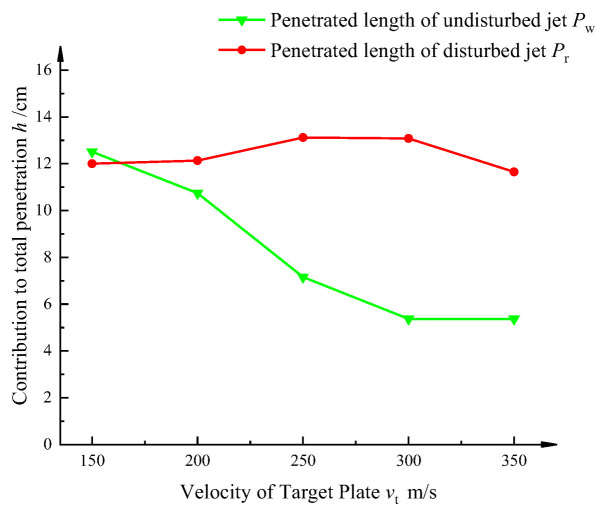
Curve of jet penetration length variation in the perturbed and unperturbed profiles.

**Figure 11 materials-15-07329-f011:**
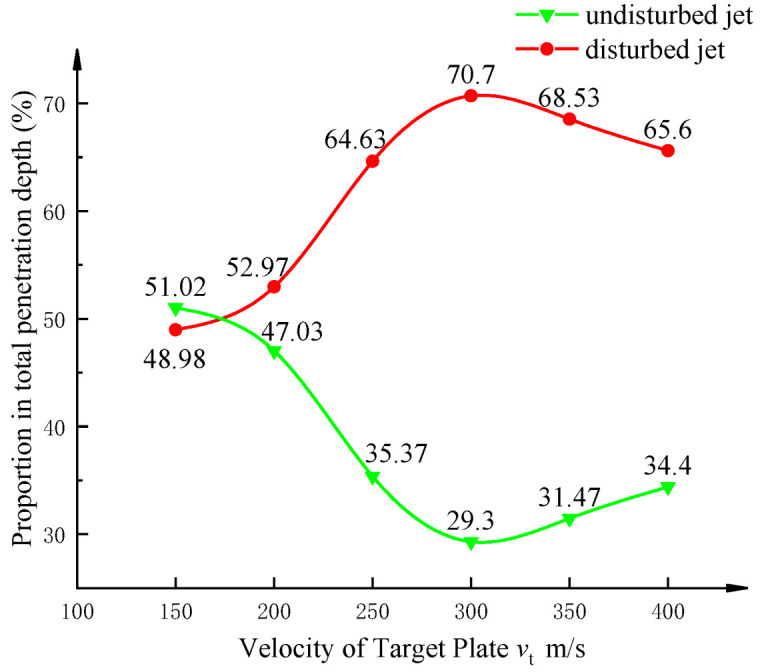
Curve of penetration length ratio of jet in perturbed and unperturbed sections.

**Table 1 materials-15-07329-t001:** Explosive Material Parameters.

*ρ*/(g·cm^−3^)	*D*/(m·s^−1^)	*A*	*B*	*R*1	*R*2	*ω*
1.891	0.911	7.783	0.0707	4.2	1	0.3

**Table 2 materials-15-07329-t002:** Material Parameters of Copper and 45 Steel.

Material	*ρ*/(g·cm^−3^)	*A/*Mbar	*B/*Mbar	*N*	*C*
copper	8.96	0.0009	0.0292	0.31	0.025
45 steel	7.85	0.00507	0.0032	0.28	0.064

**Table 3 materials-15-07329-t003:** Comparison of jet penetration depth between theoretical calculations and numerical simulations.

Target Plate Speed *v*_t_ (m/s)	Theory Calculated Value *P*_j_/cm	Numerical Value Simulation Value *P*_f_/cm	Relative Error *e*_r_ (%)
150	24.5	24.3	0.8
200	22.9	22.4	2.2
250	20.3	20.1	1.0
300	18.5	18.6	0.5
350	17.0	17.3	1.7
400	15.6	16.9	7.7

## Data Availability

Not applicable.

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
