# Peer review of "Study on Penetration Mechanism of Shaped-Charge Jet under Dynamic Conditions"

_materials, 2022, doi:10.3390/ma15207329_

Round 1
Reviewer 1 Report
Author presented mathematical model for the penetration of a jet for dynamic conditions. The work is good and has practical relevance. However, following points need to be revisied/justified before accepting the manuscript.
1) The novelty of the work must be highlighted in abstract too, which is mising right now.
2) Why the speed of 150-400 m/s is selected, add justification in result. also add related backup literature evidences in the introduction section.
3) It is also expected to mention critical review summary mentioned the potential gap at the end of introduction section.
4) Add citation to fig 1, if it is taken from literature.
5) add domain description and meshing details, if used, during simulation
6) add detailed Boundary condition description
7) Is there any available literature for "2.2 Analysis of the jet diameter in static conditions", if yes, add and compare results
8) Add version for LS-DYNA finite element analysis software. also add its availability, source, etc
9) Why undistributed and distributed results differ so much (fig 9 and 10)? recommanded to add prooper justification for the same.
10) add nomencluture
11) conclusion is missing the future scope. also i dont find any major novelty points mentioned here as well.
Author Response
Thank you for your detailed and insightful comments. I provided a point-by-point response to your comments in attachment. Please see the attachment.

Reviewer 2 Report
Abstract
The authors give a mathematical model for the penetration of a jet under dynamical conditions based on the theory of virtual origin. The penetration depth variation of shaped charge jet vertically penetrating target plates with different moving speeds is analyzed by finite element software. Also the results are compared with those of the theoretical model. Moreover the authors give some analysis in the penetration depth. It is shown that the theory can efficiently compute the jet dynamic penetration law with the target motion.
Overwiev
The topic and results are interesting additionally the paper is well-written and its language is fluent. It does includes a comprehensive literature review and the theories are given in detail. It has some minor grammatical errors and typos. I suggest the authors to read it carefully, again. Additionally, at the some references, the authors write “etc” after the names, I did not understand why it is written. Just to your take your attention. As an example: “Zhao X, Xu YJ, Zheng NN etc. Simulation and Optimization Research of Penetration Performance of Liner [J]. Journal of Ordnance 419 Equipment Engineering, 2021, 42(10): 65-71.”
Author Response
Thank you for your kind words and insightful comments. I provided a point-by-point response to your comments in the attachment. Please see the attachment.

Reviewer 3 Report
- In line 110, correct: ... model is shown in Figure"."2.
- In line 138, correct: ... increases, as shown in Figure"." 3, because the ...
- In Figure 3 would be interesting to put the measurements (in meters) in the horizontal and vertical directions; The same for Figures 4 and 5;
- In lines 243-244, the authors cite that they used a "three-dimensional model", however, it is important to present the 3D geometry with respective dimensions and an example of the mesh used;
- For the comparison criterion, it is very important that Figure 7 be presents the measurements (horizontal and vertical); the way it is, for me, it's not possible to do a clear comparison between results;
Author Response
Thank you for your comments and professional opinions on this article. I provided a point-by-point response to your comments, please see the attachments.

Round 2
Reviewer 1 Report
revision is accepted in current form.